# Self-Healing Anti-Atomic-Oxygen Phosphorus-Containing Polyimide Film via Molecular Level Incorporation of Nanocage Trisilanolphenyl POSS: Preparation and Characterization

**DOI:** 10.3390/polym11061013

**Published:** 2019-06-07

**Authors:** Bohan Wu, Yan Zhang, Dayong Yang, Yanbin Yang, Qiang Yu, Li Che, Jingang Liu

**Affiliations:** 1Space Materials and Structure Protection Division, Beijing Institute of Spacecraft Environment Engineering, Beijing 100094, China; bohanwoo@163.com (B.W.); yanbinyang_cast@163.com (Y.Y.); 13426373533@139.com (Q.Y.); 2Beijing Key Laboratory of Materials Utilization of Nonmetallic Minerals and Solid Wastes, National Laboratory of Mineral Materials, School of Materials Science and Technology, China University of Geosciences, Beijing 100083, China; 2103170021@cugb.edu.cn (Y.Z.); 17379275046@163.com (D.Y.); 3School of Science, Dalian Maritime University, Dalian 116027, China; liche@dlmu.edu.cn

**Keywords:** polyimide film, atomic oxygen, self-healing, diphenylphosphine oxide, POSS, optical properties

## Abstract

Protection of polymeric materials from the atomic oxygen erosion in low-earth orbit spacecrafts has become one of the most important research topics in aerospace science. In the current research, a series of novel organic/inorganic nanocomposite films with excellent atomic oxygen (AO) resistance are prepared from the phosphorous-containing polyimide (FPI) matrix and trisilanolphenyl polyhedral oligomeric silsesquioxane (TSP–POSS) additive. The PI matrix derived from 2,2’-bis(3,4-dicarboxyphenyl)hexafluoropropane dianhydride (6FDA) and 2,5-bis[(4-amino- phenoxy)phenyl]diphenylphosphine oxide (BADPO) itself possesses the self-healing feature in AO environment. Incorporation of TSP–POSS further enhances the AO resistance of the FPI/TSP composite films via a Si–P synergic effect. Meanwhile, the thermal stability of the pristine film is maintained. The FPI-25 composite film with a 25 wt % loading of TSP–POSS in the FPI matrix exhibits an AO erosion yield of 3.1 × 10^−26^ cm^3^/atom after an AO attack of 4.0 × 10^20^ atoms/cm^2^, which is only 5.8% and 1.0% that of pristine FPI-0 film (6FDA-BADPO) and PI-ref (PMDA-ODA) film derived from 1,2,4,5-pyromellitic anhydride (PMDA) and 4,4’-oxydianline (ODA), respectively. Inert phosphorous and silicon-containing passivation layers are observed at the surface of films during AO exposure.

## 1. Introduction

Protection of spacecraft components from long-term atomic oxygen (AO) erosion has become one of the most important issues to be addressed in the design and manufacturing of spacecraft components operating in low earth orbit (LEO) [1,2,3,4,5]. Generally, most of the organic and polymeric materials used for LEO spacecraft usually suffer from the AO attack due to the induced high impacting energy (~5.0 eV ≈ 481 kJ/mol) during the operation at an orbital velocity of ~8.0 km/s, which is higher than most of the common chemical bond dissociation energies (BDE) for polymer materials, such as C–C (~357 kJ/mol), C–H (~413 kJ/mol), C=C (~579 kJ/mol) and so on [6]. Thus, chemical bond cleavages in the molecular chains of the polymer materials will frequently occur when the materials are attacked by the high-flux AO [5]. Several well-known chemical bonds with high dissociation energies, such as Si–O, Zr–O, Ti–O, and Al–O bonds are often incorporated into the polymers to improve the AO resistance of the polymer matrixes. However, the methodologies for incorporation of these inorganic metal oxide either by physically blending as additives or surface modification as coatings inevitably deteriorate the optical properties of the derived composite films. In addition, insufficient protection from AO erosion usually occurs for the current AO resistant polymer systems due to the inner or surface deficiency caused by the blending or coating techniques [7,8,9]. 

In the past decades, AO-resistant polymers with a “self-healing” feature have attracted attention due to their abilities to automatically repair the surface damage caused by the AO erosion [10,11,12,13]. Such kind of polymers is characterized by the incorporation of specific elements in their molecular structures, such as silicon, phosphorus, titanium, and zirconium elements. These components could react with AO in-situ to form inert passivation layers, which could protect the under layers from further AO erosion. By these procedures, various intrinsically AO resistant polymers, such as polysiloxaneimide [14], phosphorus-containing poly(arylene ether benzimidazole) [15], phosphorus-containing polyimide [16], and zirconium-containing polyimides [17] have been reported in the literature. However, incorporation of such components usually compromise the optical, thermal, and mechanical properties of the pristine polymer matrixes. Consequently, more efficient polymer systems with enhanced AO resistance and good optical and thermal properties are highly desired both from the point of academic research and for practical applications. Based on the current research status of the AO-resistant polymers mentioned above, both of the “passive protection” (coating or blending AO resistant components in polymers) and “active protection” (incorporation of self-healing components) have their own advantage and disadvantages in the protective effects of AO erosion. Naturally, novel polymer systems combining both of the “passive protection” and “active protection” mechanisms might be anticipated to afford superior AO resistance to the polymer films. To the best our knowledge, few works have been reported in the literature up to now [18,19,20]. 

Recently, polyhedral oligomeric silsesquioxane (POSS) compounds have attracted attention for the R&D of AO resistant polymers due to their intrinsic nano-scale molecular structure and semi-inorganic cage Si-O-Si molecular skeleton [21,22]. The organic-inorganic interacting nature of POSS compounds makes them good candidates for high-performance components for advanced polymers. Actually, as a versatile polymer modifier, POSS has been widely investigated to improve the processability of hard-processing polymers, enhance the flame retardancy of combustible polymers, reduce the dielectric constants of standard polymers, or endow the common polymers special functionalities, such as AO resistance and so on [23,24,25]. As for the POSS modified AO resistant polymers, POSS components have been successfully incorporated into various polymer matrixes, including epoxy [26], silicone resin [27], cyanate ester resin [28], polyvinylidene fluoride (PVDF) [29], polyimide (PI) [30], polybenzoxazole (PBO) [31], and polybenzoxazine [32] so as to improve their servicing life in AO environments. Various classes of POSS-containing polymer materials with good AO resistance, including films [33], fibers [34], composites [35], aerogels [36], and so on have been developed in recent years. Among the various POSS modified AO resistant polymers, POSS-polyimide composite films have been paid much attention due to the ever-increasing requirement for high-performance PI films in LEO applications [37]. It is well known that common wholly aromatic PI films, known as Kapton^®^ film commercialized in the 1960s by DuPont, USA, are subject to atomic oxygen erosion in LEO environments [38]. In addition, the intrinsic deep color from brown to yellow of standard PI films limits their applications as optical components for LEO spacecraft. In our previous work, an AO-resistant PI film (FPI) with good self-healing features in AO environments and optical transparency was developed from fluoro-containing dianhydride, 2,2’-bis(3,4-dicarboxyphenyl)- hexafluoropropane dianhydride (6FDA) and phosphorus-containing diamine, 2,5-bis[(4-aminophenoxy)phenyl]diphenylphosphine oxide (BADPO) [39]. The FPI film exhibited good self-healing features in AO environments and good thermal stability and optical transparency. However, the relatively lower content of phosphorus shortens its lifetime when exposed to AO radiation.

In the current work, the further enhancement of AO stability of the intrinsically AO-resistant FPI film was endeavored via incorporation of trisilanolphenyl POSS (TSP–POSS) additives. The influence of the incorporation of TSP–POSS into the FPI matrix on the thermal, optical, and AO erosion features of the composite films (FPI/TSP–POSS) was investigated in detail.

## 2. Materials and Methods

### 2.1. Materials

2,5-Bis[(4-aminophenoxy)phenyl]diphenylphosphine oxide (BADPO) was synthesized in our laboratory, according to our previously reported procedure [39]. 2,2’-Bis(3,4-dicarboxyphenyl)- hexafluoropropane dianhydride (6FDA) was purchased from Tokyo Chemical Industry (Tokyo, Japan) and dried in vacuo at 180 °C for 12 h prior to use. Trisilanolphenyl-POSS (TSP–POSS) was purchased from Hybrid Plastics, Co. Ltd. (Hattiesburg, MS, USA) (Product code: SO1458) and used as received. Aminopropylisobutyl POSS (APB-POSS) was purchased from Hybrid Plastics Co. Ltd. (Hattiesburg, Mississippi, USA) (Product code: AM0265) and used as received. *N*-[(heptaisobutyl-POSS)propyl]- 3,5-diaminobenzamide (DABA-POSS) was synthesized in our laboratory. *N*-methyl-2-pyrrolidinone (NMP), *N,N*-dimethylacetamide (DMAc), *N*, *N*-dimethylforamide (DMF) were distilled prior to use and stored under a 4 Å molecular sieve. The other commercially available reagents including *m*-cresol, tetrahydrofuran, and chloroform (Yili Fine Chemicals Co. Ltd., Beijing, China) were used as received.

### 2.2. Measurements

Inherent viscosity was measured using an Ubbelohde viscometer (Brookfield Ametek, Middleboro, Massachusetts, USA) with a 0.5 g/dL PAA solution in NMP at 25 °C. Fourier transform infrared (FT-IR) spectra were measured on a Bruker Tensor-27 FT-IR spectrometer (Ettlingen, Germany). Wide-angle X-ray diffraction was conducted on a Rigaku D/max-2500 X-ray diffractometer (Tokyo, Japan) with Cu-Kα1 radiation, operated at 40 kV and 200 mA. X-ray photoelectron spectroscopy (XPS) data were obtained with an ESCALab220i-XL electron spectrometer (Thermo Fisher Scientific, Waltham, MA, USA) using 300 W of MgKα radiation. The base pressure was 3 × 10^−9^ mbar. The binding energies were referenced to the C1s line at 284.8 eV from the adventitious carbon. Field emission scanning electron microscopy (FE-SEM) was carried out using a Technex Lab Tiny-SEM 1540 (Tokyo, Japan) with an accelerating voltage of 15 KV for imaging. Pt/Pd was sputtered on each film in advance of the SEM measurements. Atomic force microscopy (AFM) was performed on a Bruker Multimode 8 AFM microscope (Santa Barbara, California, USA) in tapping mode. Ultraviolet-visible (UV-Vis) spectra were recorded on a Hitachi U-3210 spectrophotometer (Tokyo, Japan) at room temperature. The cutoff wavelength was defined as the point where the transmittance reached zero in the spectrum. Prior to the tests, PI samples were dried at 100 °C for 1 h to remove any absorbed moisture. Yellow index (YI) and haze values of the PI films were measured using an X-rite color i7 spectrophotometer (Grand Rapids, MI, USA) with PI film samples at a thickness of 20 µm. The color parameters were calculated according to a CIE Lab equation. *L** is the lightness, where 100 means white and 0 implies black. A positive *a** means a red color, and a negative one indicates a green color. A positive *b** means a yellow color, and a negative one indicates a blue color. 

Thermogravimetric analysis (TGA) was performed on a TA-Q50 thermal analysis system (New Castle, DL, USA) at a heating rate of 20 °C/min in nitrogen or in air atmosphere with the gas flowing rate of 20 mL/min. The PI film samples with the mass of 10–15 mg were used for the TGA measurements in the temperature range of 50–760 °C. Differential scanning calorimetry (DSC) was carried on a TA-Q 100 thermal analysis system (New Castle, DL, USA) at a heating rate of 10 °C/min in nitrogen with the gas flowing rate of 20 mL/min. Thermo-mechanical analysis (TMA) was recorded on a TA-Q400 thermal analysis system (New Castle, DL, USA) in nitrogen atmosphere at a heating rate of 10 °C/min with the gas flowing rate of 20 mL/min. The size of the film samples was 10 × 5 × 0.025 mm^3^. The coefficients of linear thermal expansion (CTE) values of composite films were recorded in the range of 50–200 °C. The tensile properties were performed on an Instron 3365 Tensile Apparatus (Norwood, MA, USA) with 80 × 10 × 0.05 mm^3^ samples in accordance with GB/T 1040.3–2006 at a drawing rate of 2.0 mm/min. At least six test samples were tested for each PI film and results were averaged.

The refractive index of the PI film formed on a three-inch silicon wafer was measured at the wavelength of 1310 nm at room temperature with a Metricon MODEL 2010/M prism coupler (Pennington, NJ, USA). The in-plane (*n_TE_*) and out-of-plane (*n_TM_*) refractive index were determined using linearly polarized laser light parallel (transverse electric, TE) and perpendicular (transverse magnetic, TM) polarizations to the film plane, respectively. The average refractive index (*n_AV_*) was calculated according to Equation (1):
(1)$$n_{AV}=\left(2n_{\mathrm{TE}}+n_{TM}\right)/3$$

The atomic oxygen (AO) exposure experiments were tested in a ground-based AO effects simulation facility in BISEE (Beijing Institute of Spacecraft Environment Engineering, Beijing, China), as shown in Figure 1. The facility at BISEE has three main components: An ECR plasma reactor, a neutralizer plate, and a vacuum system. The oxygen plasma is generated by the ECR plasma reactor. The magnetically confined plasma interacts with the metal plate that is negatively biased, during which oxygen ions are neutralized into oxygen atoms and are accelerated. Then the oxygen atoms are reflected from the surface toward the sample holder. The AO beam is a mixture of ions, oxygen atoms, and other species, in proportions that have not been characterized. The AO flux is characterized by the mass loss of reference Kapton exposed to AO. The facility produces an AO flux at the magnitude of 10^15^ atoms/cm^2^/s. The average kinetic energy of AO beam falls in the range of 3~8 eV, when referenced to the result of a similar device by Princeton Plasma Physics Laboratory (PPPL) [40]. Electromagnetic radiation is another byproduct caused by the dissociation and ionization along with the plasma generation. 

The AO exposure was performed on square FPI film samples with a size of 20 (length) × 20 (width) × 0.05 (thickness) mm^3^. The films were exposed to AO at a fluence of 4.0 × 10^20^ atoms/cm^2^, and the mass loss was determined. The erosion yield of the sample, *E_s_*, is calculated through the following Equation (2) [41]:
(2)$$E_{s}=\frac{\Delta Ms}{A_{s}\rho_{s}F}$$
where, *E_s_* = erosion yield of the sample (cm^3^/atom), Δ*M*_s_ = mass loss of the sample (g), *A_s_* = surface area of the sample exposed to atomic oxygen attack (cm^2^), *ρ_s_* = density of the sample (g/cm^3^), F = atomic oxygen fluence (atoms/cm^2^). As Kapton film has a well-characterized erosion yield, that is 3.0 × 10^−24^ cm^3^/atom [42], and all the present PI samples are supposed to possess similar densities and exposed area with Kapton in the AO attacking experiments, and the *E_s_* of the PIs can, therefore, be calculated using a simplified Equation (3):(3)$$Es=\frac{\Delta M_{s}}{\Delta M_{Kapton}}E_{Kapton}$$
where, *E_Kapton_* stands for the erosion yield of Kapton standard, that is 3.0 × 10^−24^ cm^3^/atom, Δ*M_Kapton_* stands for the mass loss of Kapton standard.

### 2.3. Synthesis of FPI/TSP–POSS Resins and Preparation of Composite Films

The FPI/TSP–POSS composite resin solutions were prepared according to the procedure mentioned below, which is illustrated by the synthesis of FPI/TSP-5 with a 5 wt % loading of TSP–POSS. Into a 500-mL three-necked round-bottom flask equipped with a mechanical stirrer, an ice-cold bath, and a nitrogen inlet was added BADPO (49.2500 g, 100 mmol), TSP–POSS (4.9302 g, 5.29 mmol) and newly-distilled DMAc solvent (300 g). After stirring at 5–10 °C for 30 min under a nitrogen flow, a clear diamine and TSP–POSS mixed solution was obtained. Then, 6FDA (44.4240 g, 100 mmol) was added in three batches in one hour, followed by addition of DMAc (149 g) to adjust the solid content of the mixture to be 18 wt %. The ice bath was removed and the reaction mixture was stirred at room temperature. A pale-yellow and clear viscous solution was obtained after stirring for 24 h. Then, the solution was diluted to 15 wt % by adding additional DMAc (110 g). The obtained poly(amic acid) (PAA)/TSP-5 solution was purified by filtration through a 0.45-µm polytetrafluoroethylene (PTFE) syringe filter, the obtained PAA/TSP-5 solution was stored in a brown glass bottle filled with dry nitrogen and sealed in a refrigerator at −18 °C before use.

Then, FPI/TSP-5 film was prepared as follows. The frozen PAA/TSP-5 solution was naturally warmed to room temperature in 12 h before the film fabrication. The solution was spin-coated on a silicon wafer with a diameter of 7.62 cm. The thickness of the resulting PI film was controlled by adjusting the spinning rate. The thickness of the film specimens for the FT-IR and UV-Vis measurements was controlled to be 10 µm and 25 µm, respectively, and those for thermal and mechanical property measurement were adjusted to 30–50 µm. FPI/TSP-5 film was obtained by thermally curing the PAA/TSP-5 solution under nitrogen in an oven for 1h each at 80, 150, 250, and 300 °C, respectively. Then, the free-standing FPI/TSP-5 film was obtained by immersing the silicon wafer in deionized water.

The other PI films, including FPI/TSP-10, FPI/TSP-15, FPI/TSP-20, and FPI/TSP-25 were prepared according to a similar procedure except that different TSP–POSS loadings were used. The pristine FPI film without TSP–POSS additive was also prepared with a similar procedure.

## 3. Results and Discussion

### 3.1. Evaluation and Choice of POSS Additives for PI Composite Films

One of the main purposes of this work is to develop PI composite films with long-term AO stability while maintaining good intrinsic optical transparency and thermal resistance. Thus, the choice of POSS additives is critical for achieving this target. The POSS candidates should first possess good miscibility and also comparable or superior thermal stability to the FPI matrix in order to achieve good comprehensive properties for the derived PI composite films. Thus, in our research, the solubility of three POSS candidates in the good solvent for FPI (*N*,*N*-dimethylacetamide, DMAc) and their thermal decomposition behaviors were evaluated first.

Scheme 1 shows the chemical structures of the three POSS candidates, including two commercially available POSS compounds, TSP–POSS and APB-POSS, and one home-made POSS monomer, DABA-POSS. All of the selected three POSS additives are functional and have reactive -OH or -NH_2_ groups, which are beneficial for increasing their miscibility with the poly(amic acid) (PAA) or PI matrix. The thermal stability of the POSS candidates was evaluated and Figure 2 shows the thermal decomposition behaviors of the compounds by TGA measurements both in nitrogen and in air atmospheres. Good thermal stability over 300 °C is required because the final imidization temperature is usually 300 °C in order to achieve a high degree of imidization from PAA to PI. It can be clearly observed from Figure 2 that the APB-POSS showed about 10% weight loss at 300 °C both in nitrogen and in air, while the other two compounds maintained their original weights at the same temperature. Thus, APB-POSS was abandoned due to its lower thermal stability to the other counterparts. In addition, it can be deduced from the thermal decomposition behaviours of the POSS that all of the compounds had much higher residues at the high temperature of 700 °C in air than those in nitrogen, which is in good agreement with the literature [43,44]. For example, the three POSS additives had residual weight ratios of 75.1% for TSP–POSS, 49.8% for APB-POSS, and 39.8% for DABA-POSS at 700 °C in air, respectively, which are much higher than those measured in nitrogen (TSP–POSS: 45.8%, APB-POSS: 0.6%, and DABA-POSS: 2.1%). This might mainly be due to the gain of weight from the air via the reaction of Si species in POSS and O_2_ in air. Then, the inert and thermal stable SiO_x_ formed at elevated temperature increased the residual weight ratios of the POSS compounds.

Secondly, the solubility of the TSP–POSS and DABA-POSS in DMAc at the solid content of 10 wt% was investigated. It was found that TSP–POSS exhibited good solubility in DMAc even at the solid contents as high as 30 wt%. This might be due to the pendant polar hydroxyl (-OH) groups in TSP–POSS, which are beneficial for improving the miscibility with the polar DMAc solvent. However, DABA-POSS showed poor solubility in the same tested conditions, although it has been successfully used for the synthesis of copolyimide films [37]. Polymerization with dianhydride monomers could increase its solubility in the reaction mixture. However, the poor solubility in DMAc limits its applications as an additive for PI composite films. 

In summary, TSP–POSS exhibited good thermal stability and solubility in good solvents for PAA or PI. This is mainly due to the partially condensed molecular structure of the compounds. The lateral hydroxyl groups undoubtedly increased its miscibility with various polymer matrixes.

### 3.2. FPI/TSP–POSS Composite Film Preparation

In view of the good miscibility of TSP–POSS additive with the PAA matrix, a series of PAA resins with different TSP–POSS loadings (0–25 wt % in total of the solids) were first prepared via a physical blending process, followed by thermal imidization from 80 °C to 300 °C in nitrogen, as shown in Scheme 2. The diamine and TSP–POSS were first dissolved in the solvent DMAc and 6FDA was then added incrementally. With the increase of viscosity of the reaction mixture, the system maintained a homogeneous and clear appearance from the beginning to the end of the synthesis until a viscous, pale-yellow and transparent PAA/TSP composite varnishes were obtained, as illustrated in Figure 3. The uniform nature of the PAA/TSP composite solutions indicated the molecular-level combination of the TSP–POSS additive and the PAA matrix, which is critical for obtaining the PI composite films with good optical properties.

A total of six PI composite films, FPI-0, FPI-5, FPI-10, FPI-15, FPI-20, and FPI-25 were prepared by thermally dehydrating the corresponding PAA/TSP precursors at elevated temperatures from 80 °C to 300 °C. Flexible and tough PI composite films were obtained for all of the polymers. The successful incorporation of TSP–POSS in the FPI matrix could be proven by various measurements. First, Figure 4 shows the FT-IR spectra of PI films, in which the characteristic absorptions of Si-O-Si in TSP–POSS additives at 1118 cm^−1^ appeared from absolutely nothing in FPI-0 to having a significant presence in the composite films. What is more, the strength of the Si-O-Si absorption peaks gradually increased with increasing amounts of the TSP–POSS in the composite films, indicating the molecular-level incorporation of the additives, which is in good agreement with the literature [45]. The characteristic absorption peaks of imide units labelled with the asterisks, including those at about 1778 cm^−1^ (asymmetrical stretching vibration of C=O in imides), 1721 cm^−1^ (symmetrical stretching vibration of C=O in imides), and 1372 cm^−1^ (stretching vibration of C-N in imides) were all clearly observed for the PI films, indicating the successful preparation of the polymers with the anticipated chemical structures.

Secondly, the XRD plots of the PI films are shown in Figure 5. The crystalline absorption peaks of the TSP–POSS additive completely disappeared in the spectra of the composite films, revealing the good miscibility of the TSP–POSS additive with the PI matrix.

Thirdly, the XPS spectra of the PI films were measured and Figure 6 shows the change of various elements absorptions in pristine FPI-0 and PI composite films. One clearly observes the changes of Si2p in the XPS spectra. With increasing TSP–POSS loading, the characteristic absorption peaks of Si2p appeared gradually, showing the features from Si-poor to Si-rich in the films. Meanwhile, the characteristic absorptions of C, N, O, P, and F were maintained in all of the samples, only with varying changing trends in absorption strength. 

In summary, the various measurements mentioned above indicate the successful preparation of the organic/inorganic composite films with a uniform dispersion of the TSP–POSS additive in the FPI matrix. The success is mainly attributed to the excellent solubility of the TSP–POSS fillers in the film systems. Flexible and tough PI composite films cast from the PAA/TSP precursors at elevated temperatures under nitrogen were used for the evaluation of thermal, optical, and AO erosion properties.

### 3.3. Thermal Properties

The thermal decomposition and glass transition behavior of the PI composite films were evaluated by thermogravimetric analysis (TGA) and differential scanning calorimetry (DSC) measurements, respectively, and the thermal data are summarized in Table 1. Figure 7 depicts the TGA plots of the PI films, for which the weight change of the PI films with the increasing test temperatures in nitrogen was recorded. All the films exhibited good thermal stability up to 450–500 °C and the 5% weight loss temperatures (*T*_5%_) occurred around 492–514 °C. With increasing temperatures, the films began to decompose and left higher than 50% of their original weights at 760 °C. Interestingly, it can be clearly observed from the plots that the residual weight ratios (*R*_w760_) of the PI films tend to decrease with the order of FPI-0 > FPI-5 > FPI-10 ≈ FPI-15 ≈ FPI-20 ≈ FPI-25. This trend is quite different from the PI systems based on poly(pyromellitic dianhydride-co-4,4’-oxydianiline) or poly(biphenyl dianhydride-co-*p*- phenylenediamine) and TSP–POSS additives in our previous work [46]. The *R*_w760_ values of the latter PI/TSP composite films usually increased with increasing TSP–POSS loadings because the Si–O–Si components in TSP–POSS additives can often form inert and thermo-stable residues at high temperature. However, the opposite phenomenon was observed in the current FPI/TSP–POSS systems. It might be due to the chemical reaction between the silicon-containing components in TSP–POSS and the fluorine-containing components in FPI matrix to form gaseous substance at high temperature, resulting in loss of residual weights. In our other endeavor using fluoro-containing PI derived from 6FDA and 2,2’-bis[(4-aminophenoxy)phenyl]hexafluoropropane (BDAF) as the matrix and TSP–POSS as the additive, similar phenomena were found. The detailed thermal decomposition mechanism will be investigated in future work.

Although the incorporation of TSP–POSS additives clearly affected the thermal decomposition behaviors of the PI composite films, it showed little influence on the glass transition features of the films, as can be deduced from the DSC curves shown in Figure 8. The pristine FPI-0 film exhibited a glass transition temperature (*T*_g_) of 264.2 °C, which is nearly the same as that of the other composite films. This result implies that the TSP–POSS filler does not participate in the glass transition behavior of the composite films. Although there are reactive hydroxyl groups in the molecular structure of the TSP–POSS, there should be no significant interactions between the POSS additive and the FPI matrix.

The high-temperature dimensional stability of the PI composite films was evaluated by TMA measurements and the plots are shown in Figure 9. Incorporation of TSP–POSS fillers slightly reduced the dimensional stability of the composite films at elevated temperatures. FPI-25 with a TSP–POSS loading of 25 wt % showed a coefficient of thermal expansion (CTE) of 76.3 × 10^−6^/K, which was nearly 10 × 10^−6^/K higher than that of the pristine FPI-0 film (CTE = 66.6 × 10^−6^/K). The TSP–POSS filler might play a role of “plasticizer” in the composite films, making it prone to deformation at elevated temperatures. Nevertheless, the PI composite films maintained good dimensional stability in the temperature range of 50–200 °C, which is advantageous for their applications in LEO environments.

### 3.4. Optical Properties

One of the most challenging issues for developing optically clear composite films is the uniform dispersion of the fillers. Poor dispersion of the additives will inevitably deteriorate the optical properties of the composite films. 

In the current research, because both of the TSP–POSS filler and the FPI matrix were soluble in the solvent, a molecular-level combination was achieved for the composite films. The optical properties of the PI composite films are tabulated in Table 2 and Figure 10 depicts the UV-Vis spectra of the PI films. Similar optical transmitting behaviors were observed for all of the films, indicating slight effects of the incorporation of TSP–POSS additives on the optical transmittances of the films. The PI films exhibited cutoff wavelength (*λ*_cutoff_) values below 345.0 nm and transmittances higher than 75% at the wavelength of 450 nm (*T*_450_), indicating the essentially pale-color and transparent nature of the composite films. The optical transparency of the PI films was slightly improved by the introduction of TSP–POSS when the added amounts were below 20 wt %. Higher TSP–POSS loading might deteriorate the optical transmittance of the composite films. For example, FPI-25 exhibited the relatively lowest *T*_450_ value, which might be due to the absorption of visible light by the multi-phenyl substituents in the TSP–POSS molecular structures. 

The optical transparency of the PI films was further quantitatively analyzed with the CIE Lab parameters measurements and the results were illustrated in Figure 11. The effects of filler addition on the optical properties of composite films showed some regularity. First, the yellowness indices (*b**) and haze of the composite films increases first and then decreases. For example, the haze value of the film increased from 0.72 for FPI-0 to 11.56 for FPI-5, and then gradually decreased to 3.28 for FPI-25. This might be due to the fact that when the addition of TSP–POSS was low, it acted as a dispersed phase in the film matrix. Thus, increasing the haze values of the composite films. However, when the contents of TSP–POSS increased, a continuous TSP–POSS phase gradually formed in the film matrix. As a result, the haze values of the composite films gradually decreased. 

The transition from the dispersed phase to the continuous phase for TSP–POSS fillers in the composite films could be quantitatively deduced from the change of surface roughness detected by AFM measurements, together with the labeled maximum profile peak-to-valley depth (*R*_t_) of the films, as shown in Figure 12. *R*_t_ value represents the vertical distance between the highest and lowest points in the evaluated area and describes the overall roughness of the surface. It can be clearly seen that the surface roughness of the composite films began to increase with increasing TSP–POSS filler content, especially when the surface roughness reached 15%. For example, the *R*_t_ values of the composite films changed from the initial of −9.3~8.5 nm for FPI-0 to the −46.0~74.4 nm for FPI-15. Then, with the increasing of the contents of TSP–POSS fillers in the composite films, the *R*_t_ values of the composite films decreased gradually. Finally, the FPI-25 composite films showed an *R*_t_ value in the range of −17.9~24.0 nm. This phenomenon indicates that the distribution of TSP–POSS additive in the composite films changed from dispersed phase to continuous phase with an increase of their contents. The formation of the continuous phase of TSP–POSS could be proven by the energy dispersive spectrometer (EDS) measurements of the FPI-25 film, as shown in Figure 13. On the surface of the FPI-25 film, the absorptions of C, O, and Si elements were clearly detected. The other elements, including P and F elements in the FPI matrix, were not clearly observed, although they were detectable in XPS measurements (Figure 6). 

The refractive indices (*n*) of the PI composite films are shown in Table 2. It can be concluded from the average *n* (*n_AV_*) data that the influence of the TSP–POSS fillers on the *n* values of the PI films was closely related to the chemical structure characteristics of the TSP–POSS. On one hand, TSP–POSS possesses nanocage structure, which, according to the Lorentz–Lorenz equation, is beneficial for decreasing the *n* values of the PI films due to the relatively high molar volumes of the Si-O-Si cage structures [47]. However, TSP–POSS also has three pendant polar hydroxyl groups per molecule with high molar refractions, which might increase the *n_AV_* values of the PI films to some extent. Therefore, from the point of chemical structure, the incorporation of TSP–POSS is somewhat contradictory to the effects of the refractive indices of the composite films. As shown in Table 2, when the loading of TSP–POSS was low (FPI-5), the *n_AV_* value increased slightly from 1.6229 to 1.6311, possibly caused by the polar hydroxyl groups in the TSP–POSS and the trace amounts of water they adsorbed. With increased loading of TSP–POSS, the *n_AV_* values of the composite films showed a tendency of continued decrease. This is mainly due to the domination of nanocage structures of TSP–POSS for affecting the refractive indices of the films. Finally, incorporation of TSP–POSS additives increased the birefringence of the optical PI films from the initial 0.0003 of FPI-0 to 0.0063–0.0094 for the composite films. This might be due to the locally increased degree of ordered microstructures caused by the TSP–POSS nanoparticles, which could increase the birefringence of the composite films [48].

### 3.5. AO Erosion Properties

One of the main purposes of the current work is to further enhance the long-term AO-resistant features of the intrinsically AO-resistant phosphorus-containing FPI film by incorporation of POSS additives. The “self-healing” or “self-passivating” protecting mechanisms of the intrinsic AO-resistant PI films, such as silicon, phosphorus, aluminum, and zirconium-containing PIs are well established in the literature. Inert silicates, phosphates, aluminates or zirconates in-situ form onto the surface of the functional PI films once they were attacked by the AO. Thus, further erosion of the under-layer PI films might be alleviated or postponed in AO environments. In the current research, based on the AO protecting mechanisms mentioned above, the AO-resistant ability of the FPI (6FDA-BADPO) film was further enhanced via the Si–P synergistic effects. For this purpose, the methodology of combining the “active protection” (phosphorous-containing PI matrix) and “passive protection” (TSP–POSS additives) was adopted.

The AO erosion yields (*E*_s_) of the PI films together with the FE-SEM images of FPI-0 and FPI-25 after AO exposure are shown in Figure 14 and the numerical mass lost and *E*_s_ results calculated according to the Equation (3) are tabulated in Table 3. The PI films were exposed to an AO environment with a total fluence of 4.02 × 10^20^ atoms/cm^2^. It can be seen that the *E*_s_ values of the POSS-containing composite films are much lower than that of the pristine FPI-0 film. FPI-25 film containing the highest proportion of TSP–POSS fillers showed the lowest *E*_s_ value of 0.31 × 10^−25^ cm^3^/atom, which was nearly 1.0% and 5.8% of those for the PI-ref (PMDA-ODA) and FPI-0, respectively. Clearly, incorporation of TSP–POSS additive greatly enhanced the AO resistance of the FPI films. After AO exposure, the essentially low-color and transparent PI films became somewhat opaque, which can be quantitatively deduced from the optical transmittances of the AO-eroded films at the wavelength of 450 nm (*T*_450AO_), shown in Table 2. The *T*_450AO_ values of the PI films at the thickness of 25 µm were in the range of 5.9%–58.9%, which are much lower than those of the original samples. The deterioration of the optical transparency of the PI composite films after AO erosion might be attributed to the formation of passivation layers with poor optical transmittance. This suggestion was proven by the FE-SEM images of FPI-0 and FPI-25 films after AO exposure, as illustrated in Figure 13. Compact and uniform passivation layers were found for all of the PI films eroded by AO exposure. 

The formation of passivation layers for the PI films after AO exposure could also be demonstrated by investigating the surface roughness of the AO-attacked surface of the films. The AFM images of a 2D height scan on an area of 10 µm × 10 µm for AO-eroded PI films are shown in Figure 15. When the TSP–POSS content is within 15 wt% in the composite films, exposure to AO greatly increased the roughness of the PI sample surfaces. However, when the TSP–POSS content was higher than 15 wt%, the roughness of the PI surfaces apparently decreased. The FPI-25-AO composite film showed the lowest *R*_t_ value in the range of −5.8~5.5 nm after AO exposure and a continuous and dense passivation layer was clearly observed from the AFM image. It can be inferred that the formation of the surface passivation layer has a good protective effect on the underlying polymer.

On the other hand, by comparing the surface morphology of the AO-eroded PI films shown in Figure 14, it is seen that for FPI-0 without TSP–POSS filler, a single passivation layer consists of uniform peaks and valleys with phosphorus components formed on its surface after AO exposure (which could be verified by EDS measurements). However, for the FPI-25 film containing 25 wt % of TSP–POSS, more than one kind of passivation layer formed on the surface after the AO attack. Besides the dense passivation layer similar to that of FPI-0-AO, irregular passivating structures could also be clearly observed. This indicates that the AO erosion mechanism of PI composite films containing both phosphorus and silicon elements might be different from that of the traditional single-protective PI materials. In order to investigate the effects of the TSP–POSS additive and the phosphorus on the AO durability of the PI films, the surface chemistry of the PI samples after AO exposure was also studied. Figure 16 shows the high-resolution XPS spectra of Si2p and P2p for unexposed and exposed PI films and the relative atomic concentration of the unexposed and exposed films are listed in Table 4. As shown in the table, the relative atomic concentration of O, Si, and P significantly increased after AO exposure, whereas that of the C decreased dramatically. The Si2p binding energy peaks shifted from the initial 102.5 eV of the pristine samples to 103.4 eV of the AO-eroded samples, indicating the occurrence of an oxidation reaction for the silicon elements. The P2p absorption peaks of PI films shifted from binding energy of 132.4 eV to a higher value of 133.4 eV. It can be concluded that silicon oxides (or silicate) and phosphorus oxides (or phosphate) layers formed on the surface of the PI films after AO exposure. In addition, some composition characteristics of the FPI/TSP composite film could be deduced from the XPS measurements. For the unexposed samples, the atomic concentration of Si elements in the surface compositions of the PI films increased with increasing TSP–POSS loadings. However, the P elements could not be detected any more when the TSP–POSS loadings were higher than 15 wt %. A lower P concentration than anticipated has also been observed in our previous study [39], which might be due to the molecular chain morphography of the diphenylphosphine oxide units in the PIs. It can be further deduced from the XPS results that the passivation layer formed on the surface of PI films after AO exposure is composed more of silicon-containing substances than phosphorus-containing components. Moreover, the higher the contents of TSP–POSS fillers in the composite film structure, the higher the silicon content in the surface passivation layer. The above phenomenon also corresponds to the erosion yield of the PI films in the AO environment. 

## 4. Conclusions

In the current work, the intrinsically AO-resistant phosphorus-containing PI film, FPI (6FDA-BADPO) was used as a host for the nanocage TSP–POSS additives so as to further enhance the long-term AO stability of the composite films while maintaining the intrinsic good properties of the matrix. This was successfully achieved according to the various measurements. First, TSP–POSS showed excellent miscibility with the FPI matrix and a molecular-level combination could be achieved. Secondly, incorporation of TSP–POSS did not deteriorate the thermal and optical properties of the pristine film. Thirdly, the AO resistance of the FPI matrix film was significantly enhanced via the Si–P synergic effect. The FPI-25 composite film exhibited an AO erosion yield of 0.31 × 10^−25^ cm^3^/atom after an AO attack with a fluence of 4.02 × 10^20^ atoms/cm^2^, which is only 5.8% and 1% that of pristine FPI-0 film and PI-ref (PMDA-ODA), respectively. The XPS and FE-SEM measurements revealed a double AO passivation mechanism of active-protecting of phosphorus components and the passive-protecting of silicon constituents. This double-protecting procedure might endow the PI composite films with excellent long-term stability in AO environments.

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
