# Peer review of "Self-Healing Anti-Atomic-Oxygen Phosphorus-Containing Polyimide Film via Molecular Level Incorporation of Nanocage Trisilanolphenyl POSS: Preparation and Characterization"

_polymers, 2019, doi:10.3390/polym11061013_

Round 1

Reviewer 1 Report

In this work the authors prepared and characterized a series of novel organic/inorganic polyimide-based nanocomposite with different functionalized POSS for a possible application in low-earth orbit spacecraft.

The manuscript is well written and the Introduction quite well descriptive of the state of the art (probably can be enriched with some references, see attached .pdf). If is possible the authors have to try to abbreviate the title. In the subsection measurements please indicate roughly the mass of the sample used for TGA and the range of investigation as well as if the atmosphere is static or flowing (in this case also the flowing rate). In addition, in the experimental the authors described the atmosphere as inert but in the results and discussion they declared that the tests were performed also in oxidative condition, please clarify this point. In Results and discussion some statements can be strengthened by supporting them with literature references (see attached .pdf). In Figs 7, 9 and 10 I suggest the removing of the points from the curves and their differentiation by using colors, so as to make the comparison more readable.

Anyway, the discussion is interesting in agreement with the conclusion that absolutely deserve publication. Some minor changes are suggested (see attached .pdf) before acceptance for publication.

Author Response

1.     With respect to the suggested references, we added them in our revised manuscript as follows.

“To the best our knowledge, few works have been reported in the literature up to now [18-20].

Recently, polyhedral oligomeric silsesquioxane (POSS) compounds have attracted attention for the R&D of AO resistant polymers due to their intrinsic nano-scale molecular structure and semi-inorganic cage Si-O-Si molecular skeleton [21,22].”

2.     With respect to the title of the manuscript, we shortened it in our revised manuscript as follows.

Self-Healing Anti-Atomic-Oxygen Phosphorus-Containing Polyimide Film via Molecular Level Incorporation of Nanocage Trisilanolphenyl POSS: Preparation and Characterization”.

3.     With respect to the TGA measurements, we added the details in our revised manuscript as follows.

Thermogravimetric analysis (TGA) was performed on a TA-Q50 thermal analysis system (Delaware, USA) at a heating rate of 20 oC/min in nitrogen or in air atmosphere with the gas flowing rate of 20 mL/min. The PI film samples with the mass of 10-15 mg were used for the TGA measurements in the temperature range of 50-760 oC.”.

4.     With respect to the atmospheres for TGA measurements, we added the expression in our revised manuscript as follows.

Thermogravimetric analysis (TGA) was performed on a TA-Q50 thermal analysis system (Delaware, USA) at a heating rate of 20 oC/min in nitrogen or in air atmosphere with the gas flowing rate of 20 mL/min.”.

5.     With respect to the format of curves in Figure 7, Figure 9, and Figure 10, we removed the points from the curves in our revised manuscript as follows.

Figure 7. TGA curves FPI/TSP films at a heating rate of 20 oC/min under nitrogen flow.

Figure 9. TMA curves FPI-TSP films at a heating rate of 10 oC/min under nitrogen flow.

Figure 10. UV-Vis spectra of FPI-TSP films.

6.     With respect to the revisions shown in the attached file, we all modified them in our revised manuscript as shown in the Revision list. 

Reviewer 2 Report

In this manuscript by Bohan Wu et al., the authors report the protection of polymeric materials, i.e phosphorous-containing polyimide (FPI) matrix and TSP-POSS additives, from atomic oxygen erosion. To understand the role of TSP-POSS additives, they measured and analyzed various chemical/physical/optical properties. The properties are consistent and support their conclusion. However, there are some concerns to be clearly addressed to be publishable (see below). I recommend the publication of this manuscript after minor revision.

In Fig. 6 XPS data, the peak intensities of F1s and Si2P are not seem to be consistent to the amount of TSP-POSS. In case of FPI-20, F1s peak is very small comparing to others. In addition, the Si2P peak intensities does not linearly increased or decreased. But, XPS of Fig. 15 shows linearly increased intensity of Si2P peak as a function of the concentration of TSP-POSS. Could you comment the disagreement of Fig. 6 and 15?

In Fig. 13, SEM images shows the surface state of films after exposure. To understand the AO erosion, it is required to compare the SEM images before and after exposure. In addition, AFM images of Fig. 12 and 14 are also difficult to understand due to different image scale. To compare the affect after exposure, it is required to compare AFM images with before exposure with same image scale.

To understand the optical properties (haze properties), they suggest a continuous TSP-POSS phase at high concentration of TSP-POSS. To support the continuous TSP-POSS phase, they mentioned large roughness from AFM. But, roughness from AFM cannot be support the formation of continuous phase. To support the continuous phase formation, EDS mapping or other data is needed.

Author Response

1.     With respect to the disagreement of peak intensities of F1s and Si2p in Fig. 6 and Fig. 15, we thought that although XPS measurements could reflect the elemental composition of the PI film surface quantitatively and qualitatively, the analysis results are highly affected by the surface composition of the samples, especially for composites. For composites, the elemental composition in the different surface position might be slightly different. Thus, in the XPS measurements, different peak intensity results might be obtained.

In the current research, TSP-POSS were physically blended with the PI matrix with a loading amount from 0 to 25%. XPS measurements were used to analyze the film surface qualitatively. More precise and quantitative results might be difficult to be obtained due to the limitation of the preparation and dispersion technique in the work. For Figure 6 and Figure 15, different samples were used for XPS measurements. Thus, the change of intensity of Si2p peak might be slightly different in these two figures.

2.     With respect to the comparison of SEM images of PI samples before and after AO exposure, we added the SEM image of FPI-25 before AO exposure in Figure 13 in our revised manuscript.

3.     With respect to the comparison of AFM images of PI samples before and after AO exposure, we change Figure 14 with a new figure with the same image scale (2um) in our revised manuscript as follows.

Figure 14. The 2D AFM images (10 µm × 10 µm) of PI films after AO exposure (4.0×1020 atom/cm2).

4.       With respect to the characterization of continuous TSP-POSS phase, we measured the EDS of FPI-25 film and added them in our revised manuscript as follows.

Figure 13. EDS images of FPI-25 film.

The corresponding discussion was added as follows.

The formation of the continuous phase of TSP-POSS could be proven by the energy dispersive spectrometer (EDS) measurements of the FPI-25 film, as shown in Figure 13. On the surface of the FPI-25 film, the absorptions of C, O, and Si elements were clearly detected. The other elements, including P and F elements in the FPI matrix were not clearly observed, although they were detectable in XPS measurements (Figure 6).

Revision list

1.     Page 392, at the end of sentence, three new sentences were added as follows.

The formation of the continuous phase of TSP-POSS could be proven by the energy dispersive spectrometer (EDS) measurements of the FPI-25 film, as shown in Figure 13. On the surface of the FPI-25 film, the absorptions of C, O, and Si elements were clearly detected. The other elements, including P and F elements in the FPI matrix were not clearly observed, although they were detectable in XPS measurements (Figure 6).”.

2.     Page 394, a new figure was added as Figure 13.
